# Controversy about the Role of Rifampin in Biofilm Infections: Is It Justified?

**DOI:** 10.3390/antibiotics10020165

**Published:** 2021-02-05

**Authors:** Nora Renz, Andrej Trampuz, Werner Zimmerli

**Affiliations:** 1Center for Musculoskeletal Surgery, Charité-Universitätsmedizin, Corporate Member of Freie Universität Berlin, Humboldt-Universität zu Berlin, and Berlin Institute of Health, 10117 Berlin, Germany; nora.renz@charite.de; 2Department of Infectious Diseases, Bern University Hospital, University of Bern, 3010 Bern, Switzerland; 3Interdisciplinary Unit of Orthopaedic Infections, Kantonsspital Baselland, 4410 Liestal, Switzerland; werner.zimmerli@unibas.ch

**Keywords:** rifampin, biofilm, prosthetic joint infection

## Abstract

Rifampin is a potent antibiotic against staphylococcal implant-associated infections. In the absence of implants, current data suggest against the use of rifampin combinations. In the past decades, abundant preclinical and clinical evidence has accumulated supporting its role in biofilm-related infections.In the present article, experimental data from animal models of foreign-body infections and clinical trials are reviewed. The risk for emergence of rifampin resistance and multiple drug interactions are emphasized. A recent randomized controlled trial (RCT) showing no beneficial effect of rifampin in patients with acute staphylococcal periprosthetic joint infection treated with prosthesis retention is critically reviewed and data interpreted. Given the existing strong evidence demonstrating the benefit of rifampin, the conduction of an adequately powered RCT with appropriate definitions and interventions would probably not comply with ethical standards.

## 1. Introduction

Rifampin is one of the first-line drugs against tuberculosis. In addition, it has been used against non-mycobacterial microorganisms, mainly staphylococci, for at least 50 years [1]. However, its place in severe staphylococcal infections not involving an implanted device remained unclear for decades because no systematic comparative studies had been performed. In the meantime, few studies have been published on this topic. In five randomized controlled trials and two retrospective cohort studies in patients with *Staphylococcus aureus* bacteremia, no difference of mortality could be shown [2]. A recent multicenter, randomized, double-blind placebo-controlled trial confirmed these data in 758 patients [3]. In the study of Rieg et al. [4], only the subgroup of patients with implants had less late complications related to *S. aureus* bacteremia when treated with combination therapy (4.5% vs. 10.6%, *p* = 0.03). Most of them were treated with a rifampin combination regimen, suggesting a benefit of antibiofilm activity compared to treatment without rifampin. In contrast, the addition of rifampin to standard therapy showed no advantage in patients with native valve infective endocarditis caused by *S. aureus* [5]. Thus, the latest data advocate against the uncritical use of rifampin combination therapy in patients with severe staphylococcal infections in absence of implants.

In contrast, the benefit of rifampin in patients with staphylococcal implant-associated infection is well documented based on abundant in-vitro, animal, and clinical data, as summarized in a recent review [6]. Until recently, only one randomized controlled trial (RCT) existed, in which the added value of rifampin was shown in patients with orthopedic implant-associated staphylococcal infections [7]. In 2020, a second RCT in patients with periprosthetic joint infection (PJI) was published, using different combination therapy regimens, which did not show a better outcome with addition of rifampin to standard treatment [8]. These unexpected data may unsettle clinicians with limited experience in the field of implant-associated infections. Therefore, possible reasons for the failure of demonstrating the benefit of adding rifampin in this trial will be discussed herein in the light of available evidence, including animal data and clinical trials. 

## 2. Short History of Rifampin Use in Patients with Implant-Associated Staphylococcal Infection

In 1982, the use of rifampin in the treatment of non-tuberculous infections has been initially presented in a large symposium, followed by the publication in a supplemental edition of the Reviews of Infectious Diseases, edited by Merle A. Sande [9]. The special interest in rifampin was based on its unique mode of action, i.e., its inactivation of the bacterial DNA-dependent RNA polymerase. Its main drawback is the single-step mutation of the rifampin-binding enzyme occurring with a frequency of 10^−^^6^ to 10^−^^7^ [10]. This high risk of emergence of resistance explains its occasional failure in infections characterized by a high bacterial load, such as in infective endocarditis or persistent *S. aureus* bacteremia [5,11,12]. Studies of rifampin in non-mycobacterial infection were retarded by the fear that its widespread use could result in resistance to rifampin in *Mycobacterium tuberculosis*. 

One of the first observations of the successful use of rifampin combination therapy in implant-associated infections is the report of two patients with *S. epidermidis* infection, one with prosthetic valve endocarditis and the other with ventriculoperitoneal shunt-associated infection [13]. In a case series, Karchmer et al. [14] reported a good outcome with a vancomycin-rifampin, but not betalactam–rifampin combination (87% vs. 43%, *p* = 0.025) for treatment of prosthetic valve endocarditis caused by methicillin-resistant *S. epidermidis*. These data suggest that the combination partner of rifampin matters.

Based on our observation that rifampin could not only prevent, but also cure experimental staphylococcal implant-associated infections [15], we performed additional animal experiments with rifampin combination therapy [16], followed by observational studies and one randomized controlled trial in patients with orthopedic implant-associated infections [7,17,18,19]. Later, rifampin combination therapy has shown to improve the outcome in patients with other types of implant-associated infections such as staphylococcal prosthetic valve endocarditis [14,20], deep sternal wound infections [21] and vascular graft associated infections [22,23]. However, data from randomized controlled trials are still not available in patients with non-orthopedic implant-associated infections.

## 3. Evidence for the Efficacy of Rifampin in Animal Studies

The first observation of the biofilm activity of rifampin has been made >35 years ago in the guinea pig tissue cage model [15]. With four doses of rifampin, implant-associated *S. aureus* infection could be cured in 100% of the tissue cages, if therapy was started up to 12 h after inoculation. If the delay was prolonged to 24 h, the cure rate decreased to 57%. These results unequivocally demonstrate that rifampin is able to eliminate surface-adhering biofilm staphylococci. However, it also shows that the efficacy of a short-term therapy is limited to a young biofilm. A clear definition of the limit between young and mature (tolerant) biofilm is still lacking. It depends on the microorganism, the antibiotic, and the duration of therapy [24]. Table 1 summarizes several experimental studies with the subcutaneous tissue cage animal model in guinea pigs. In each experimental series, rifampin combinations were significantly more active than other antibiotics [16,25,26,27,28,29,30]. This animal model does not simulate orthopedic device-related infection. However, it allows following an ongoing infection with the most relevant endpoint, namely complete elimination of the biofilm. Other groups investigated the role of rifampin in animal models of implant-associated osteomyelitis, and corroborated the antibiofilm effect of rifampin, as summarized in a recent review [6].

## 4. Role of Rifampin in Clinical Studies Involving Orthopedic Implant-Associated Infections 

Based on the animal data showing an impressive antibiofilm activity of rifampin against staphylococci, we started to treat patients with orthopedic device-related infection (ODRI) with rifampin combination in clinical routine. In a first case series, 10 patients with staphylococcal ODRI undergoing debridement and implant retention (DAIR), the success rate was 80% [17]. In this and many subsequent studies, no direct comparison is possible, because either none or all patients were treated with rifampin combinations. In patients treated with DAIR without rifampin combination therapy, the success rates were as low as 31% to 35% [31,32]. However, in these studies, the Infectious Diseases Society of America (IDSA) guidelines regarding the indication for DAIR have not been considered [33]. 

In the study of Holmberg et al. [34], patients with staphylococcal knee PJI had a better failure-free survival, when treated with a rifampin combination than without rifampin (81% vs. 41%, *p* = 0.01). Similarly, in a study from the Mayo Clinic, patients treated with DAIR according to the IDSA-guidelines including a rifampin-regimen had a better outcome than patients in a historical control group treated without rifampin (93% vs. 63%) [35]. However, in this study, most of the patients received long-term suppressive antimicrobial therapy. 

In several studies, all patients undergoing DAIR for staphylococcal PJI were treated with a rifampin-regimen. The failure-free survival ranged between 80% and 100% in patients treated according to the IDSA-guidelines, in whom the rifampin combination could be given for a prolonged time (generally >2 months) [36,37,38,39,40,41,42,43]. In a study, in which 29 patients with acute PJI were treated with ciprofloxacin plus rifampin, the success rate was 83% [39]. Interestingly, in the mentioned Norwegian randomized trial, in which rifampin-combination therapy did not show superiority, another regimen has been used, namely cloxacillin or vancomycin with or without rifampin [8]. Possible reasons for the low success rates and the lack of improvement by the addition of rifampin are presented below. Indeed, diligent choice of antimicrobial agents may be crucial. In the observational study of Puhto et al. [44] in patients with PJI treated with DAIR, treatment success was significantly higher in patients with ciprofloxacin/rifampin as compared to those with another combination partner or a regimen without rifampin.

Despite the overwhelming evidence for the antibiofilm activity of rifampin, there are a few studies, in which no beneficial effect of rifampin was shown. Bouaziz et al. [45] showed that non-compliance with IDSA guidelines was a risk factor for treatment failure in patients with hip or knee PJI. However, rifampin as single factor was not advantageous because of the strong association between surgical therapy and outcome. Thus, rifampin combination therapy should only be used in patients qualifying for DAIR [33,46]. In an observational study of patients with acute PJI treated with DAIR and linezolid with or without rifampin, patients receiving rifampin did not have an improved outcome. The confounder in this study may be the high prevalence of polymicrobial infection in both groups (41% and 35%, respectively) indicating that many patients may have had wound healing disturbance or even a sinus tract during therapy [47]. 

Rifampin long-term therapy is complicated by its frequent gastrointestinal side effects, and its strong induction of isoenzymes of cytochrome P450 [6,10]. This is a major clinical challenge, as the effect of rifampin can only be considered in patients in whom it can be given for a sufficient duration. Enzyme induction by rifampin leading to drug-drug interactions requires specific attention prior to and at the end of treatment. However, the interaction of rifampin and other antibiotics in vitro is difficult to interpret, because synergism/antagonism in vitro does not correlate with the effect in vivo [48]. Based on experimental data, the antibiofilm effect seems to be a class effect of all rifamycin derivatives [26,49,50]. First clinical data suggest that rifabutin is a valuable alternative to rifampin with less adverse events and less drug-drug interactions [51].

## 5. Critical Appraisal of a Randomized Controlled Trial (RCT) Showing no Effect of Rifampin

The above mentioned RCT compared the outcome of patients with acute staphylococcal PJI treated with prosthesis retention and either monotherapy without rifampin or rifampin combination [8]. In this multicenter study conducted from 2006 to 2012 in eight centers, 48 patients with acute PJI were included in the final analysis. PJI was caused by methicillin-susceptible staphylococci in 38 episodes (among them 36 were *S. aureus*) and 10 by methicillin-resistant staphylococci (of which all were *S. epidermidis*). Twenty-five patients were randomized to receive monotherapy, i.e., cloxacillin (two weeks intravenous, followed by four weeks oral) or vancomycin (six weeks intravenous) and 23 patients received rifampin in addition to the anti-staphylococcal treatment regimen mentioned above. 

All patients underwent “soft tissue” revision with retention of the prosthesis. Re-revision with isolation of any pathogen was considered confirmed failure, while clinical signs of infection without revision surgery or isolation of pathogen were categorized as probable failure. Using the Kaplan–Meier method, the infection-free survival rate was similar in the monotherapy group (72%) and rifampin combination group (74%) at two years follow-up (median, 27 months). Success rate in PJI caused by methicillin-susceptible staphylococci was 78% with rifampin combination and 65% with monotherapy. In PJI caused by methicillin-resistant staphylococci, monotherapy was successful in all five patients (100%), whereas rifampin-vancomycin-combination had a success of 60% (three of five). No statistically significant difference was observed in any comparison. The authors conclude that adding rifampin to standard antibiotic treatment in acute staphylococcal PJIs does not improve the outcome.

In view of the above presented role of rifampin as biofilm-active antibiotic, the results of this RCT unsettled clinicians with limited experience in the field. Therefore, some critical points in this study should be highlighted for correct interpretation of the results.

First, the originally registered study protocol at ClinicalTrials.gov (NCT00423982) differs from the published manuscript, suggesting that relevant modifications were performed during the study. In contrast to the initial protocol, in addition to patients with early postoperative PJI those with acute hematogenous PJI were included. In late hematogenous PJI, the duration of infection is less well defined, because it may manifest only delayed after seeding. This may explain that the success rate of PJI treated with DAIR has shown to be significantly lower in late acute staphylococcal infection as compared to early postoperative infections [52]. Unfortunately, the distribution of the two clinical entities in the analyzed cohort is not provided, making the interpretation of the results of the heterogeneous study population difficult. 

Second, the surgical treatment is described in the Methods in detail. Whereas in the trial registration protocol, only a “soft tissue” revision is mentioned, in the manuscript additionally exchange of modular parts, irrigation with 9 L of saline and placement of two gentamicin-containing sponges (10 × 10 cm^2^) is stated, exceeding the procedure of a soft tissue revision. The adherence to this strict surgical protocol throughout the six-year study in eight study centers is questionable, as inclusion in the study took place most likely only after identification of the causing pathogen. Exchange of mobile parts being a proxy for a thorough debridement was shown to be among most relevant factors for successful outcome in several previous studies in case of retained infected prosthesis [36,53,54,55]. Noteworthy, no dropouts due to deviating surgical treatment were reported. 

Third, the antimicrobial combination partner for rifampin is crucial, as mentioned by the authors in the Discussion. In this study, unusual combinations with oral cloxacillin (low oral bioavailability (37%), poor bone penetration, low maximal dose orally compared to intravenous route [56]) and prolonged intravenous vancomycin (toxic, poorly penetrating into the bone, barely bactericidal, non-therapeutic levels upon initiation of treatment) in case of methicillin-resistance were administered. Substances recommended as antimicrobial combination partner for rifampin are those with a high oral bioavailability and a good bone penetration, such as quinolones, trimethoprim-sulfamethoxazole, doxycycline or clindamycin, none of which was used in the present study. In addition, an unusual rifampin dosage (300 mg three times daily) was used, which is neither approved nor recommended for any indication. 

Fourth, the absence of infectious diseases specialists in the author list suggests lack of an interdisciplinary team approach to the management of PJI, which is another important factor determining the treatment success of PJI [57,58]. After discharge, adequate intake or administration of antibiotics, patient compliance and modification in case of intolerance should be ensured. Rifampin is often discontinued due to intolerance or toxicity, as shown by the high number of dropouts (n = 7) due to rifampin discontinuation in this study. The accompaniment by an infectious diseases specialist during the treatment period could probably counteract the high dropout rate and potential selection bias. 

Fifth, probably the most relevant drawback of the study is the low number of included patients. The final analysis with 48 patients in eight centers during six years indicates a reluctant recruitment. Since staphylococci are the most frequent pathogens of acute PJI [59,60], the average of one patient per center per year implies that the participating centers are not explicitly centers specialized in septic surgery and that the included patients represent a subgroup of patients bearing the risk of selection bias.

Sixth, due to the low number of included subjects, the study is underpowered, and thus does not allow any conclusion on the effect of rifampin on the outcome of acute staphylococcal PJI. The sample size calculation required at least 62 patients in each group to statistically prove an increase in cure rate of 20% (assuming a high cure rate of 70% in the monotherapy group). The authors aimed to include at least 100 subjects in each group. Only focusing on methicillin-susceptible staphylococci, the success rate with monotherapy was 65% (13 out of 20 patients), whereas the rifampin combination led to treatment success in 78% (14 out of 18 patients). Based on theoretical considerations, by increasing the number sample size sixfold (120 patients in the monotherapy group, 108 patients in the combination group) and assuming the same proportion of success in each group, the results would reach statistical significance. Unfortunately, the study was prematurely stopped without mentioning the reason for discontinuation. Only by increasing the sample size the beneficial effect of rifampin could have probably been shown, if there is one, as suggested by multiple above-mentioned studies. 

Finally, there are a few imprecisions regarding the outcome evaluation, the reader should consider while interpreting the study results. It remains unclear to what extend the “probable” failures were true septic failures. Furthermore, it is not indicated, whether non-microbiological criteria (synovial fluid leukocyte count and periprosthetic tissue histopathology) for infection were fulfilled in these cases. In addition, the meticulous analysis of failures to discriminate relapse or infection caused by a new pathogen (superinfection) is missing, however, of utmost importance. The fact that the study was conducted several years ago would have allowed for assessment of long-term follow-up. However, only two-year follow-up was reported. Taking all these aspects into consideration, the discussed study does not allow any deduction on the effect of rifampin on the outcome of acute staphylococcal PJI treated with DAIR.

## 6. Conclusions

Taken together, the controversy about the role of rifampin in biofilm infections is not justified. There is abundant data from in-vitro and animal experiments, as well as clinical studies confirming its antibiofilm effect in patients with staphylococcal orthopedic implant-associated infections undergoing DAIR. Thus, one study with multiple weaknesses should not unsettle clinicians. An RCT with appropriate sample size, optimal choice of antimicrobials, standardized surgical interventions and accurate definition of treatment failure would be desirable. However, given the existing strong evidence demonstrating the benefit of rifampin, the conduction of such a clinical study would not comply with ethical standards and would probably not be approved by ethics committees.

## Figures and Tables

**Table 1 antibiotics-10-00165-t001:** Cure rate in the guinea pig tissue cage infection model (copyright© American Society for Microbiology, Antimicrob Agents Chemother 63(2), e01746-18, 2019 [6]).

Microorganism	Antibiotic Regime	Cure Rate	*p ^a^*	Reference
*S. epidermidis* B3972 (clinical strain)	CiprofloxacinCiprofloxacin + Rifampin	0%100%			<0.01	Widmer et al. 1990 [16]
*S. aureus* ATCC 29,213 (MSSA)	VancomycinVancomycin + Rifampin	0%75%			<0.01	Zimmerli et al. 1994 [25]
CiprofloxacinCiprofloxacin + Rifampin	17%92%			<0.001
*S. aureus* ATCC 29,213 (MSSA)	LevofloxacinLevofloxacin + Rifampin	0%88%		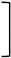	<0.001	Trampuz et al. 2007 [26]
Levofloxacin + ABI-0043 ^b^	92%	
*S. aureus* ATCC 43,300 (MRSA)	LinezolidLinezolid + Rifampin	0%60%		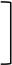	<0.001	Baldoni et al. 2009 [27]
Levofloxacin + Rifampin	91%	
*S. aureus* ATCC 43,300 (MRSA)	DaptomycinDaptomycin + Rifampin	0%67%		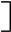	<0.001	John et al. 2009 [28]
*S. aureus* ATCC 43,300 (MRSA)	DalbavancinDalbavancin + Rifampin	0%36%		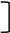	<0.001	Baldoni et al. 2013 [29]
*S. aureus* ATCC 43,300 (MRSA)	FosfomycinFosfomycin + Rifampin	0%83%			<0.001	Mihailescu et al. 2014 [30]

^a^ Fisher’s exact test for categorical variables, statistical significance is defined as *p* < 0.05. ^b^ ABI-0043 is a derivative of Rifalazil, which is a rifamycin derivative.

## Data Availability

Data is contained within the article.

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
