# Peer review of "Controversy about the Role of Rifampin in Biofilm Infections: Is It Justified?"

_antibiotics, 2021, doi:10.3390/antibiotics10020165_

Round 1

Reviewer 1 Report

Should be published in Antibiotics after minor revision

In their Perspective manuscript entitled “Controversy about the role of rifampin in biofilm infections: Is it justified?”, Nora Renz, Andrej Trampuz and Werner Zimmerli discuss the potential of rifampin for treating biofilm infections, based on a critical overview of relevant studies in preclinical and clinical settings. After ascertaining the anti-biofilm effect of rifampin, authors aim to provide an objective re-interpretation of results reported last year by Karlsen et al., who suggested no better outcome with addition of rifampin in different combination therapy regimens compared to standard treatment in patients with periprosthetic joint infection. Authors provide their expertise for convincingly stating that, when appropriately employed, rifampin is a potent antibiotic against implant associated infections and controversy about the role of that antibiotic in biofilm infection is not justified.

In my opinion, this paper is quite well-written and pleasant to read. It compiles 59 references to relevant previous studies, including some that were very recently published. Among these, 17 were authored by Zimmerli, Zimmerli and Trampuz, or Renz and Trampuz. Based on their long-standing expertise, authors aim to provide recommendation useful to clinicians for the adequate use of rifampin. For all these reasons, I recommend publication of this manuscript in Antibiotics after a minor revision considering the few comments listed below.

Minor comments

In the Abstract, the writing can be improved. The two last sentences are almost a repetition of the two last sentences in Conclusion.

Some typo errors must be corrected. For instance, in the abstract line 14: “and o clinical evidence”, delete “o”.

In Table 1, the formatting can be improved. The antibiotic regime of the first study listed in that Table is not correctly indicated. Horizontal lines may be added to better distinguish each study and improve readability. Statistical significance (P) should be defined using a footnote.

Some references to previous reports by other authors may be added, notably:

  • Dunne WM Jr, Mason EO Jr, Kaplan SL. Diffusion of rifampin and vancomycin through a Staphylococcus epidermidis biofilm. Antimicrobial Agents and Chemotherapy. 1993 Dec;37(12):2522-2526. DOI: 10.1128/aac.37.12.2522.
  • Zheng Z, Stewart PS. Penetration of rifampin through Staphylococcus epidermidis biofilms. Antimicrob Agents Chemother. 2002;46(3):900-903. doi:10.1128/aac.46.3.900-903.2002
  • Perlroth J, Kuo M, Tan J, Bayer AS, Miller LG. Adjunctive Use of Rifampin for the Treatment of Staphylococcus aureus Infections: A Systematic Review of the Literature. Arch Intern Med.2008;168(8):805–819. doi:10.1001/archinte.168.8.805
  • Hung-Jen Tang, Chi-Chung Chen, Kuo-Chen Cheng, Kuan-Ying Wu, Yi-Chung Lin, Chun-Cheng Zhang, Tzu-Chieh Weng, Wen-Liang Yu, Yu-Hsin Chiu, Han-Siong Toh, Shyh-Ren Chiang, Bo An Su, Wen-Chien Ko, Yin-Ching Chuang In Vitro Efficacies and Resistance Profiles of Rifampin-Based Combination Regimens for Biofilm-Embedded Methicillin-Resistant Staphylococcus aureus Antimicrobial Agents and Chemotherapy Oct 2013, 57 (11) 5717-5720; DOI: 10.1128/AAC.01236-13

Author Response

Point by point reply for manuscript ID antibiotics-1096605: Controversy about the role of rifampin in biofilm infections: Is it justified? by Nora Renz, Andrej Trampuz and Werner Zimmerli

Response to Reviewer 1 Comments

  1. In the Abstract, the writing can be improved. The two last sentences are almost a repetition of the two last sentences in Conclusion.

Authors’ reply: We reformulated the last two sentences (page 1, lines 21-24)

“Given the existing strong evidence demonstrating the benefit of rifampin, the conduction of an adequately powered RCT with appropriate definitions and interventions would probably not comply with ethical standards.”

  1. Some typo errors must be corrected. For instance, in the abstract line 14: “and o clinical evidence”, delete “o”.

Authors’ reply: Thank you for the comment. We deleted the “o” and corrected other typographical errors (page 1, line 14).

  1. In Table 1, the formatting can be improved. The antibiotic regime of the first study listed in that Table is not correctly indicated. Horizontal lines may be added to better distinguish each study and improve readability. Statistical significance (P) should be defined using a footnote.

Authors’ reply: We formatted the Table 1 according to your suggestions:

  1. Some references to previous reports by other authors may be added, notably:
  • Dunne WM Jr, Mason EO Jr, Kaplan SL. Diffusion of rifampin and vancomycin through a Staphylococcus epidermidis biofilm. Antimicrobial Agents and Chemotherapy. 1993 Dec;37(12):2522-2526. DOI: 10.1128/aac.37.12.2522.
  • Zheng Z, Stewart PS. Penetration of rifampin through Staphylococcus epidermidis biofilms. Antimicrob Agents Chemother. 2002;46(3):900-903. doi:10.1128/aac.46.3.900-903.2002
  • Perlroth J, Kuo M, Tan J, Bayer AS, Miller LG. Adjunctive Use of Rifampin for the Treatment of Staphylococcus aureus Infections: A Systematic Review of the Literature. Arch Intern Med.2008;168(8):805–819. doi:10.1001/archinte.168.8.805
  • Hung-Jen Tang, Chi-Chung Chen, Kuo-Chen Cheng, Kuan-Ying Wu, Yi-Chung Lin, Chun-Cheng Zhang, Tzu-Chieh Weng, Wen-Liang Yu, Yu-Hsin Chiu, Han-Siong Toh, Shyh-Ren Chiang, Bo An Su, Wen-Chien Ko, Yin-Ching Chuang In Vitro Efficacies and Resistance Profiles of Rifampin-Based Combination Regimens for Biofilm-Embedded Methicillin-Resistant Staphylococcus aureus Antimicrobial Agents and Chemotherapy Oct 2013, 57 (11) 5717-5720; DOI: 10.1128/AAC.01236-13

Authors’ reply: We did not add the publications by Dunne et al., Zheng et al, and Tang et al., because these three studies exclusively deal with in vitro experiments. In our manuscript, we focused on animal experiments and clinical data. However, we added the publication by Perlroth et al. in the paragraph on drug-drug interactions. Including this paper allowed us to mention the poor correlation of synergism/antagonism-experiments in vitro with antimicrobial effects in vivo (see page 5, lines 160-163)

Reviewer 2 Report

Line 14: There is a typographical error “o”. Please rectify.

Line 35: Bacterial scientific names (eg S. aureus and M. tuberculosis) should be in italics – this needs to be fixed throughout the manuscript

Line 90: Please change “With the delay” to “If the delay”

Table 1: In the first line there is a mistake as Ciprofloxacin is written twice, when I believe one of them is meant to be on the line underneath and written as “Ciprofloxacin + Rifampin”. This is almost an exact replica of the table cited in the table legend – have the authors sought permission from the original authors to reproduce their data?

Author Response

Point by point reply for manuscript ID antibiotics-1096605: Controversy about the role of rifampin in biofilm infections: Is it justified? by Nora Renz, Andrej Trampuz and Werner Zimmerli

Response to Reviewer 2 Comments

  1. Line 14: There is a typographical error “o”. Please rectify.

Authors’ reply: We improved the abstract and reformulated the last two sentences (page 1, lines 21-24)

  1. Line 35: Bacterial scientific names (eg S. aureus and M. tuberculosis) should be in italics – this needs to be fixed throughout the manuscript.

Authors’ reply: We changed the bacterial scientific names into italics throughout the entire manuscript.

  1. Line 90: Please change “With the delay” to “If the delay”

Authors’ reply: Done. “If the delay was prolonged to 24 hours, the cure rate decreased to 57%.” (page 3, line 94)

  1. Table 1: In the first line there is a mistake as Ciprofloxacin is written twice, when I believe one of them is meant to be on the line underneath and written as “Ciprofloxacin + Rifampin”. This is almost an exact replica of the table cited in the table legend – have the authors sought permission from the original authors to reproduce their data?

Authors’ reply: We corrected the formatting imprecisions . The table is identical to the cited version. As Werner Zimmerli is author of the respective publication in AAC, no permission is needed to publish his own table. The required citation format was respected (“Copyright © American Society for Microbiology, [insert journal name, volume number, page numbers, and year]”).
